# Increased Plasma Levels of ACE and Ang II in Prediabetes May Contribute to Adipose Tissue Dysfunction

**DOI:** 10.3390/ijms26125517

**Published:** 2025-06-09

**Authors:** Bongeka Cassandra Mkhize, Palesa Mosili, Phikelelani Sethu Ngubane, Ntethelelo Hopewell Sibiya, Andile Khathi

**Affiliations:** 1School of Laboratory Medicine & Medical Sciences, University of KwaZulu-Natal, Durban 3629, KwaZulu-Natal, South Africa; bongeka.mkhize28@gmail.com; 2Department of Human Physiology, University of KwaZulu-Natal, Westville 4000, KwaZulu-Natal, South Africa; 215065077@stu.ukzn.ac.za (P.M.); ngubanep1@ukzn.ac.za (P.S.N.); 3Department of Pharmacology, University of Rhodes, Grahamstown 4000, Eastern Cape, South Africa; n.sibiya@ru.ac.za

**Keywords:** renin–angiotensin system, adipose tissue, adiposity, metabolic dysfunction, prediabetes, type 2 diabetes

## Abstract

Adipose tissue is essential for the regulation of insulin sensitivity and cytokine production, which are key processes in maintaining metabolic homeostasis. Previous studies have shown a link between the renin–angiotensin system (RAS) and adipose tissue dysfunction in type 2 diabetes (T2D); however, the role of RAS in prediabetes remains underexplored. This study aimed to analyze the association between RAS components and adipose tissue dysfunction in the prediabetic state. This observational, cross-sectional study was conducted between 21/05/21 and 20/05/24 and analyzed RAS markers in plasma samples. This study was conducted at King Edward Hospital, focusing on individuals from outpatient clinics. The study included non-prediabetic (NPD), prediabetic (PD), and T2D individuals (n = 40 per group) aged 25–45 years. The participants were selected based on fasting blood glucose levels and HbA1c criteria. Plasma RAS markers and adipose function markers were measured in each participant. Primary outcomes included HOMA-IR, HbA1c, and plasma levels of ACE1, Ang II, ACE2, Ang 1-7, adiponectin, adipsin, MCP-1, and HDL. PD participants had significantly altered glycaemic control (HOMA-IR: 2.1 ± 0.4 vs. 3.9 ± 0.8; HbA1c: 4.9 ± 0.4 vs. 5.9 ± 0.6) compared to NPD. Plasma ACE1 (162.0 ± 10.55 vs. 180.3 ± 7.546) and Ang II (20.26 ± 2.404 vs. 25 ± 1.752) were elevated, while adiponectin (29.08 ± 5.72 vs. 23.22 ± 4.93) and HDL (1.01 ± 0.11 vs. 0.67 ± 0.11) were reduced in PD. Alterations in RAS manifest early in prediabetes and are associated with adipose tissue dysfunction. These findings suggest that RAS dysregulation contributes to early metabolic disturbances in prediabetes.

## 1. Introduction

Adipose tissue cells are regarded as endocrine organs because they facilitate multiple metabolic roles through the secretion of adipokines such as adiponectin, adipsin, and RAS axes [1,2]. In adipose tissue, there are two distinct axes, ACE1/Ang II (angiotensin-converting enzyme 1/angiotensin II) and ACE2/Ang 1-7 (angiotensin-converting enzyme 2/angiotensin 1-7) that affect adipogenesis [3]. The ACE1/Ang II arm of the RAS is associated with insulin resistance, adipose tissue inflammation, metabolic dysfunction, and adipose tissue dysfunction by modulating adipocyte differentiation [4]. Conversely, ACE2/Ang 1-7 counteracts the ACE1/Ang II arm; in mature adipocytes, it aids in the proliferation of insulin-sensitive mature adipocytes and differentiation of preadipocytes [5].

However, in T2D, the balance of these two axes is dysregulated in favour of ACE1/Ang II, consequently altering the adipokines adiponectin and adipsin [6]. The secretion and actions of these adipokines are modulated in prediabetes and T2D, subsequently promoting morphological and biochemical alterations in adipose tissues [7]. Enlarged adipocytes, which result from over-nutrition in T2D, induce inflammation in addition to a range of pro-hyperglycemic adipokines, such as monocyte chemoattractant protein-1 (MCP-1) [8,9]. These prohyperglycemic adipokines promote insulin resistance, impaired glucose tolerance, impaired fasting glucose, and elevated glycated hemoglobin levels, which are risk factors for prediabetes [10].

Prediabetes is an asymptomatic condition in which the blood glucose concentration is above the physiological range and below the threshold for the diagnosis of type 2 diabetes mellitus (T2DM) [11]. This condition is characterized by impaired fasting glucose (IFG), defined as fasting plasma glucose (FPG) of 5.6–6.9 mmol/L, impaired glucose tolerance (IGT), defined as 2 h plasma glucose of 7.8–11.0 mmol/L, and a glycated hemoglobin A1c (HbA1c) of 5.7% to 6.4% [12]. Prediabetes, which often precedes the onset of T2DM, may be regarded as an early predictor of T2DM onset [13]. Globally, the number of people with T2DM has increased from 108 million in 1980 to 463 million in 2019, which has resulted in an increase in the prevalence in adults aged >18 years from 4.7% in 1980 to 9.3% in 2019 [14]. This indicates that people in low-middle, middle, and high-middle socio-demographic index (SDI) countries, such as South Africa, are more prone to type 2 diabetes because of social and economic transformation with increased food supply, high-calorie diets, and reduced physical activity [15]. South Africa has seen a rapid increase in the prevalence of diabetes, which almost tripled from 4.5% in 2010 to 12.7% in 2019 [15]. It was estimated that of the 4.58 million people aged 20–79 years with diabetes in South Africa in 2019, 52.4% were undiagnosed, which may be because prediabetes is asymptomatic [16]. However, studies have reported biochemical alterations in tissues such as adipose tissue during the prediabetic state. In South Africa, more than 60% of women and 31% of men have adipose tissue dysfunction, with obesity being one of the major drivers of non-communicable diseases such as T2DM [17].

An animal study determined the presence of RAS in adipose tissue in prediabetes [18]. The study highlighted a positive correlation between prediabetes, upregulation of RAS, and adipose tissue dysfunction, which resulted in ectopic fat redistribution [18,19]; however, the presence of RAS in the adipose tissue of humans with prediabetes and its relationship with associated adiposity biochemical and clinical markers, such as adipsin, cytokines, and adiponectin, has not been established. While our study did not include direct measures of adiposity such as body mass index (BMI) or waist circumference, it focuses on the biochemical signatures of adipose tissue dysfunction associated with RAS dysregulation. Therefore, this study aimed to address an important translational gap: it explores the interplay between RAS activity and adipose tissue biochemistry in human prediabetes, potentially illuminating early mechanisms of metabolic deterioration prior to overt diabetes.

## 2. Results

### 2.1. Characterization Table of the Participants

#### 2.1.1. Glycated Hemoglobin and Homeostatic Model Assessment of Insulin Resistance (HOMA-IR)

The glycated hemoglobin percentages obtained from non-prediabetic (NPD), prediabetes (PD), and type 2 diabetic (T2D) individual reports were extracted for analysis. In the same groups, plasma insulin concentration and fasting blood glucose (FBG) levels were measured to calculate the Homeostatic Model Assessment of insulin resistance (HOMA-IR). The results indicated a statistically significant elevation in glycated hemoglobin, FGB, and HOMA-IR in the PD group compared to the NPD group. Furthermore, the T2DM group exhibited a significantly higher HOMA-IR and glycated hemoglobin percentage (HbA1c %) than both the NPD and PD groups.

##### Plasma Renin Concentration

Figure 1 displays the plasma renin concentrations of the NPD, PD, and T2DM groups. The results showed that the plasma concentration of renin was significantly higher in the PD and T2D groups in comparison to the NPD group. However, renin levels were lower in the PD group than in the T2D group.

##### ACE1 and Ang II

Figure 2 displays the plasma angiotensin-converting-enzyme 1 (ACE1) and angiotensin II (Ang II) concentrations of the NPD, PD, and T2D groups, which were measured. The results showed that the T2D group plasma had significantly higher concentrations of ACE1 and Ang II in comparison to the NPD and PD groups. Furthermore, the ACE1 and Ang II were significantly higher in the PD in comparison to the NPD, and lower when compared to the T2D.

##### ACE2 and Ang 1-7

Figure 3 displays the plasma (A) angiotensin-converting-enzyme 2 (ACE2) and (B) angiotensin 1-7 (Ang 1-7) concentrations of the NPD, PD, and T2D groups, which were measured. The results showed that the T2D group plasma had significantly lower concentrations of ACE2 and Ang 1-7 in comparison to the NPD and PD groups. However, the PD group had higher ACE2 and Ang 1-7 concentrations in comparison to the NPD and T2D.

##### Adipsin

Figure 4 Displays the adipsin concentrations of the NPD, PD, and T2DM groups which were measured. The PD group had a significantly higher concentration of adipsin in comparison to the NPD. The results showed that the T2D group plasma had a significantly lower concentration of adipsin in comparison to the PD and NPD groups.

##### Adiponectin

Figure 5 Displays the plasma adiponectin concentrations of the NPD, PD, and T2D groups which were measured. The results showed that the T2D group plasma had a significantly lower concentration of adiponectin in comparison to the PD and NPD groups. The PD group had a significantly lower concentration of adiponectin compared to the NPD group.

##### Monocyte Chemoattractant Protein-1 (MCP-1)

Figure 6 displays the plasma MCP-1 concentrations of the NPD, PD, and T2D groups which were measured. The results showed that the T2D group plasma had a significantly higher concentration of MCP-1 in comparison to the PD and NPD groups. The PD group had a significantly higher concentration of MCP-1 in comparison to the NPD.

## 3. Discussion

Adipose tissue serves as the main store for fat accumulation, but it also performs extensive connective tissue and endocrine activities, producing hormones that regulate several metabolic processes, most notably, glucose homeostasis [20]. This complex control is mediated by the interaction between anti-hyperglycemic hormones, namely adiponectin and adipsin, as well as the renin-angiotensin system (RAS) [21,22]. In type 2 diabetes mellitus (T2DM), hyperglycemia is associated with alterations in the RAS, leading to various complications [23]. Prediabetes is characterized by asymptomatic, chronically elevated blood glucose concentrations that precede the onset of type 2 diabetes mellitus (T2DM) [24]. Although a wealth of information exists regarding the disruptions associated with T2DM, including the aberrant activity of the RAS in various organs, notably in adipose tissue, the specific involvement of adipose RAS activity in the prediabetic stage and the underlying mechanistic pathway leading to the development of prediabetes and its progression to T2DM remains unclear [18]. In T2DM, the persistence of hyperglycaemia promotes chronic activation of the RAS, particularly within adipose tissue, where it disrupts adipocyte function, alters adipokine secretion, and fosters a pro-inflammatory environment [25,26]. Increased angiotensin II levels and AT1 receptor activity have been shown to decrease adiponectin production, promote oxidative stress, and impair insulin signalling, thereby exacerbating insulin resistance and glucose intolerance [27,28]. Adipose tissue dysfunction in T2DM is also marked by macrophage infiltration, fibrosis, and altered lipid metabolism, processes that are partially driven by elevated local RAS activity [29].

These alterations highlight not only the pathological role of adipose RAS in established T2DM but also suggest that early dysregulation of this system may precede and contribute to the transition from prediabetes to overt diabetes.

This study aimed to explore the intricate interplay between RAS activity and prediabetes. This was achieved by evaluating the various components of the RAS and their impact on adipokines, such as adipsin and adiponectin. Adipose tissue harbours the necessary components for local RAS activation, including renin, angiotensin-converting enzyme (ACE1), angiotensin II (Ang II), angiotensin-converting enzyme (ACE2), angiotensin 1-7 (Ang 1-7), aldosterone, and angiotensin receptors, thus allowing for the autonomous control of RAS within adipose depots [30].

RAS is essential for the physiological processes of adipocyte development and regulation [31]. Renin serves as a key regulatory enzyme that initiates the RAS within adipose tissue by catalyzing the conversion of angiotensinogen to angiotensin I, a critical step that influences adipocyte function and regulation [32]. In type 2 diabetes (T2D), renin levels are often elevated, promoting insulin resistance and metabolic dysfunction [33]. Renin has also been implicated in the upregulation of pro-inflammatory cytokines and the promotion of oxidative stress within adipose tissue, contributing to chronic low-grade inflammation in T2D [34]. Furthermore, elevated renin expression has been associated with increased aldosterone production, further compounding insulin resistance and impairing glucose metabolism [18]. The results of this study showed significantly increased renin levels in T2D patients (see Figure 1). Additionally, renin levels were significantly higher in prediabetic (PD) patients than in the non-prediabetic (NPD) group (see Figure 1). Renin levels subsequently influence the concentrations of Ang II and Ang 1-7, depending on the upregulation of ACE1 and ACE2, respectively [35]. ACE2/Ang 1-7 in mature adipocytes aids in the proliferation of insulin-sensitive mature adipocytes and the differentiation of preadipocytes [4,31,36]. As a result, this mechanism improves the insulin signalling pathway and its influence on triglyceride metabolism and control [34]. In T2DM, the ACE2/Ang 1-7 interaction is downregulated and ACE1/Ang II signalling is upregulated. Upregulation of the ACE1/Ang II arm of the RAS is associated with adipose tissue dysfunction by modulating adipocyte differentiation [37,38]. In T2D, the downregulation of ACE2/Ang 1-7 and upregulation of ACE1/Ang II exacerbate the impairment of insulin signalling and dysregulation of triglyceride metabolism and control [39]. This imbalance promotes chronic inflammation and endothelial dysfunction, both of which are hallmarks of T2D-associated vascular complications [40]. Increased Ang II has also been shown to impair glucose uptake in peripheral tissues by disrupting insulin receptor substrate signalling [41]. Therefore, in T2D, the downregulation of ACE2/Ang 1-7 and upregulation of ACE1/Ang II are positively correlated with an increase in HOMA-IR and decreased HDL, indicating insulin resistance and impaired triglyceride control, respectively, subsequently leading to hyperlipidemia [39]. The T2D results of the current study concur with the available literature, wherein ACE2 and Ang 1-7 were significantly lower and ACE1 and Ang II were significantly higher in the T2D group than in the non-prediabetic (NPD) group (see Figure 2 and Figure 3). These biomarkers were also measured in the prediabetic (PD) stage, which is an intermediate stage, in order to gain an understanding of the biochemical changes that occur during the development and progression of T2D. The results of this study demonstrate for the first time that biochemical changes, namely, ACE1 and Ang II, begin to be upregulated in prediabetes and progress further as the glycemic state worsens (see Figure 2).

In humans, Ang II produced by mature adipocytes inhibits the differentiation of adipocyte precursors, thereby decreasing the percentage of small insulin-sensitive adipocytes that produce adipokines [22,42]. Ang II may inhibit adipocyte differentiation partly due to its effects on insulin, which is an adipogenic hormone that may contribute to the enlargement of existing insulin-resistant adipocytes [22,43]. Physiologically, this process is regulated by the upregulation and binding of Ang II to angiotensin II type 2 receptors (AT2R) instead of angiotensin II type 1 receptors (AT1R) in the adipose tissue [31]. Additionally, Ang II is converted by ACE2 into Ang 1-7, thus regulating insulin sensitivity, glucose homeostasis, and adipokine production [22,30,44]. In T2D, due to the increase in Ang II production in conjunction with a decrease in ACE2, which converts Ang II into Ang 1-7, there is insulin resistance and a decrease in adipokine production leading to overt hyperglycemia, which is evidenced by increased levels of glycated hemoglobin (HbAlc) [45,46]. Moreover, the predominance of Ang II in T2D has been shown to suppress PPARγ expression, a key transcription factor involved in adipogenesis and insulin sensitivity [28,47]. This suppression further limits the formation of metabolically healthy adipocytes and contributes to systemic insulin resistance [48]. The T2D results of this study corroborate previous findings reported in T2D, whereby the increased Ang II and decreased Ang 1-7 concentration were directly proportional to significantly increased HOMA-IR and HbAlc indicative of insulin resistance and overt hyperglycemia, respectively, in the T2D group compared to the NPD group (see Table 1 and Table 2). High HOMA-IR and HbAlc accompanied the deranged RAS in patients with PD. This could indicate the co-existence of insulin resistance and RAS derangement in patients with PD in this study. The results of this study align with those of previous studies showing that RAS modulation in T2D is associated with insulin resistance and overt hyperglycemia. Interestingly, the PD results of this study uniquely demonstrated that RAS changes occur before the diabetes diagnosis threshold and may further contribute to the development of insulin resistance and moderate hyperglycemia in prediabetes. Studies have demonstrated that Ang II inhibits insulin-mediated PI3K pathway activation, thereby impairing endothelial NO production and Glut-4 translocation in insulin-sensitive tissues, which results in vascular and systemic insulin resistance, respectively [46]. Although we could not confirm the role of RAS in the progression of diabetes in this study, it could serve as a cornerstone for adopting RAS hyperactivation as a key pathological process in the conversion of prediabetes to overt diabetes. Indeed, evidence from the literature suggests that inhibition of RAS through ACE1 inhibitors and ARBs is attenuated. Treatment with RAAS blockers has been shown to improve parameters of glucose metabolism and decrease the incidence of new-onset diabetes in patients with or without hypertension who are at high risk for diabetes [49]. These studies, together with our observations, may further underscore the impact of the RAS on diabetes development and progression.

As mentioned, ACE1/Ang II signalling inhibits the differentiation of insulin-sensitive preadipocytes and adipocytes that produce adipokines, such as adipsin and adiponectin, which are essential for glucose homeostasis [50]. Adipsin, also known as complement factor D, is involved in an alternative pathway of the complement system [51]. Adipsin also plays a crucial role in maintaining adipose tissue homeostasis and pancreatic β-cell function [52]. Studies have shown that long-term chronic adipsin supplementation in db/db mice ameliorates hyperglycemia and increases plasma insulin concentration while preserving beta cells [53]. Furthermore, adipsin catalyzes the release of complement factor C3a, which stimulates insulin production in pancreatic β-cells [53]. Therefore, adipsin plays a protective role in beta cells, which are often deficient in T2DM patients [54]. In T2DM, adipsin is reduced, resulting in dedifferentiation and apoptosis of pancreatic β-cells, consequently exacerbating hyperglycemia to overt hyperglycemia [54,55]. The reduced adipsin levels observed in T2DM patients have also been linked to impaired insulin secretion and a diminished incretin response [55]. Moreover, lower adipsin levels are associated with heightened inflammatory signalling, further impairing metabolic regulation in individuals with T2DM [56]. Plasma adipsin concentration was found to be high in T2DM patients in the current study, which is consistent with previous reports (see Figure 4). Interestingly, PD presented with elevated adipsin, but it was lower than that in T2DM (see Figure 4). These observations are of interest, and whether progressive increases occur before the onset of T2DM remains elusive. Physiologically, adipsin plays a protective role in pancreatic β cells [40]. Conversely, in the context of T2DM, substantial depletion of adipsin levels precipitates apoptosis of β-cells, culminating in diminished insulin synthesis [38,40]. Hence, with the marked increase in ACE1/Ang II signalling, as evidenced by the elevated levels of HbAlc, HOMA-IR, and adipsin in the PD group compared to the NPD group in this study, we propose that the upregulation of ACE1/Ang II may have influenced the differentiation of insulin-sensitive preadipocytes and the maturation of adipocytes, thereby potentially reducing the population of cells capable of producing adipsin (see Table 2). These findings potentially elucidate the evolution of hyperinsulinemia in prediabetes and the development of the insufficient insulin phenomenon evident in T2DM patients. The decreased adiponectin levels in the PD group were positively correlated with glycated hemoglobin and HOMA-IR in this study, thereby elucidating the mechanistic underpinnings that contribute to the initiation and advancement of metabolic dysfunction observed during the transition from prediabetes to T2DM (see Figure 5). Moreover, adiponectin has been shown to inhibit ACE1/Ang II; therefore, due to the significantly reduced adiponectin in PD, we hypothesized that this reduction contributed to the upregulated ACE1/Ang II interaction, which was evidenced by the increased plasma ACE1 and Ang II in PD compared to NPD.

Adiponectin has also been found to be independently correlated with HDL-C levels in non-diabetic men and women [57]. The mechanism underlying this correlation may be that adiponectin enhances the secretion of apolipoprotein A-I (apo-AI), which is the major apolipoprotein of HDL [57,58,59]. Additionally, adiponectin increases the expression of the ATP-binding cassette transporter A1 (A-BCA1), which induces HDL assembly through reverse cholesterol transport in hepatic cells [59,60]. HDL plays a crucial role in lipid metabolism due to its role in reverse cholesterol transport [61]. One study demonstrated that the administration of Ang II in wild-type mice resulted in a reduction in circulating HDL levels [62,63,64]. This reduction is associated with the translocation of scavenger receptor type B1 (SR-B1) proteins to the plasma membrane within adipose tissue. Analogous findings have been observed in transgenic mice characterized by the overexpression of angiotensinogen in adipose tissue [65,66]. In addition to animal studies, clinical investigations have reported triglyceride metabolism dysfunction in T2D [67]. In T2D, HDL levels are frequently diminished, which may impair cholesterol efflux capacity and contribute to lipid accumulation in peripheral tissues [68]. Furthermore, HDL particles in T2D are often dysfunctional, exhibiting reduced anti-inflammatory and antioxidant properties, which exacerbates metabolic complications [68]. The results of the T2D group in this study align with those of studies in which HDL was significantly reduced in the T2D group compared to the NPD (see Table 1). HDL was also analyzed in prediabetes, whereby it was significantly reduced in the PD group compared to the NPD group. In this study, HDL was also significantly lower in T2D patients than in PD patients, thus indicating that as glucose levels increase, triglyceride metabolism is disrupted, resulting in triglyceride dysfunction in hyperglycemia, which is seen in T2D. Therefore, triglyceride levels were measured in T2D in this study and were found to be significantly increased in T2D patients compared to NPD, which correlates with the literature (see Table 1). Furthermore, in the current study, triglycerides (TGs) were also measured in the PD group and were found to be higher in the PD group than in the NPD group and lower in the PD group than in the T2D group. As mentioned, disruption of RAS homeostasis in favour of the ACE1/ang II arm leads to decreased adipose tissue buffering capacity and insulin resistance, consequently resulting in increased free fatty acid (FFA) plasma triglycerides [4,34]. As stated in this study, there was a progressive increase in TGs levels as glucose levels increased (see Table 1). Increased TGs and insulin resistance are known to promote hepatic cholesterol synthesis; hence, T2D increases FFA, TGs, and total cholesterol levels [69,70]. Furthermore, in T2D, the upregulation of the ACE1/Ang II arm in adipose tissue has been shown to promote triglyceride metabolism and dyslipidemia, which affects not only HDL availability, total cholesterol, and TG4s, but also causes and increases in low-density lipoproteins (LDL) and very low-density lipoproteins (VLDL) [71,72]. In the current study, HDL, TGs, total cholesterol, LDLs, and VLDLs were measured in T2D patients, whereby HDLs were decreased in the T2D group, while total cholesterol, TGs, LDLs, and VLDLs were significantly higher in T2D patients than in NPD (see Table 1). These parameters were also measured in PD patients, where the same trend of decreased HDL and decreased total cholesterol, TGs, LDLs, and VLDLs was observed in the PD group compared with the NPD (see Table 1). Therefore, we postulate that increasing glucose levels modulate RAS by upregulating the ACE1/ang II arm, consequently affecting the availability of insulin-sensitive preadipocytes and adipocytes that produce adipokines, such as adiponectin, subsequently affecting HDL bioavailability. These observations raise the hypothesis that heightened levels of Ang II originating from adipocytes may contribute to obesity-related dyslipidemia by modulating adipocyte SR-B1 expression and influencing HDL clearance. Furthermore, due to the role of adiponectin in regulating ACE1/Ang II signalling, we speculate that the reduction in adiponectin in PD compared to NPD may further contribute to the dysregulation of RAS and triglyceride control, cholesterol synthesis, and the balance between good fats such as HDL and bad fats, namely, LDL and VLDL, which are indicative of the early stages of a vicious cycle. We posit that as glucose levels worsen from PD to T2D, this cycle may contribute to the development of comorbidities associated with T2D, such as obesity.

Additionally, ACE1/Ang II signalling perpetuates lipotoxicity and the enlargement of adipocytes, thereby affecting fat mass and promoting inflammation in adipose tissue [21,73]. As fat mass expands, as reported in prediabetes and obesity, inflammation is seen in ectopic fat depots and fat-free sites [74]. The ACE1/Ang II interaction can stimulate chemokine secretion and induce increased oxidative stress, contributing to inflammation [75]. For instance, in preadipocytes extracted from rat adipose tissue, Ang II increases the expression of monocyte chemoattractant protein-1 (MCP-1), a chemokine involved in inflammatory responses. Similarly, Ang II infusion in rats increased MCP-1 expression in adipose tissues [76]. Ang II induces the expression of MCP-1 in rat preadipocytes through a pathway dependent on nuclear factor-kappa B (NF-κB) [76,77]. This was demonstrated in a study by Tsuchiya et al., where Ang II significantly increased MCP-1 mRNA levels in a time- and dose-dependent manner [76,78]. This increase was completely abolished by an Ang II type 1 (AT1)receptor antagonist, indicating the involvement of the AT1 receptor in this process [79]. In T2D, MCP-1 levels are often elevated in both serum and adipose tissue, where they contribute to chronic low-grade inflammation by promoting monocyte infiltration and macrophage activation [80]. This macrophage accumulation, particularly of the M1 phenotype, results in the secretion of pro-inflammatory cytokines, further impairing insulin receptor signalling [81]. Furthermore, studies have shown that increased MCP-1 expression in T2D patients correlates with higher HOMA-IR values, reduced insulin sensitivity, and deteriorating glycaemic control, suggesting its potential as both a biomarker and therapeutic target in managing insulin resistance and inflammation in T2D [82]. Taking into account these biochemical changes, we postulate that in prediabetes, moderate hyperglycemia alters the RAS, consequently promoting inflammation, as evidenced by the increased MCP-1 concentration in PD compared to NPD (see Figure 6). Subsequently, due to the chronic state of moderate hyperglycemia, the upregulation of ACE1/Ang II signalling persists, thus worsening insulin resistance, glucose tolerance, and inflammation, as evidenced by the significantly increased ACE1/Ang II, HOMA-IR, and MCP-1 in T2DM. Therefore, the results of this study may provide insights into the mechanism that contributes to metabolic dysfunction and the development of a plethora of derangements as prediabetes progresses to T2DM.

The ACE2/Ang 1-7 pathway plays a crucial role in regulating visceral adipose tissue expansion and ER stress [83]. ACE2, an upstream regulatory enzyme, cleaves Ang II to produce Ang 1-7, which acts mainly through the Mas receptor [84,85]. Deletion of ACE2 or Mas can affect the expression levels or action of Ang 1-7 [85]. In the context of obesity and T2DM, ACE2 deletion or Mas deletion leads to increased visceral adipose tissue, higher leptin levels, larger adipocyte size, and upregulated lipogenesis and ER stress-related proteins [43,86]. However, treatment with Ang 1-7 decreases visceral adipose tissue mass and adipose leptin, suggesting that the ACE2/Ang 1-7 pathway could be a potential therapeutic target for obesity and T2DM [33,87,88]. The ACE2/Ang 1-7 signalling, counteracts the effects of angiotensin II and is recognized as an adipokine [22,40]. It has been shown that increased activation of the ACE2/Ang 1-7/MasR arm can improve lipid profile and insulin resistance, and reduce inflammation [88,89]. Interestingly, in the current study, ACE2 and Ang 1-7 levels were higher in the PD group than in the NPD group.

In PD, moderate hyperglycemia triggers activation of the ACE2/Ang 1-7 pathway as a compensatory mechanism to counterbalance the detrimental effects of the ACE1/Ang II axis. Since ACE2/Ang 1-7 is the protective arm of the renin–angiotensin system, it initially works to offset increasing glucose levels and insulin resistance. However, with persistent hyperactivation of the ACE1/Ang II axis, along with continued hyperglycemia and insulin resistance, MasR activity begins to diminish. This impaired the protective effects of the ACE2/Ang 1-7/MasR arm. In PD, we postulate that the significantly elevated ACE1/Ang II, coupled with decreased adipokines and increased inflammation, ultimately overrides the protective effects of ACE2/Ang 1-7. The consequences of ACE1/Ang II overactivity became more evident, further contributing to metabolic dysfunction.

## 4. Methods and Materials

### 4.1. Study Design and Sampling

This study procured archived blood samples from individuals treated at King Edward VIII Hospital (KEH) in Durban, South Africa, and linked clinics in the eThekwini metropolitan area. The sample encompassed both male and female subjects aged between 25 and 45 years, representing diverse ethnic backgrounds. Exclusions were made for Individuals with a medical history of liver disease, kidney disease, heart disease, and clinical depression were excluded. Additionally, samples from expectant mothers and professional athletes were excluded from the study. The study protocol was approved by the University of Kwa Zulu-Natal Biomedical Research Ethics Committee (BREC) (reference no. BE266/19), and the Provincial Health Research Committee of the KZN Department of Health. All the participants provided informed verbal consent. This study was conducted in accordance with the ethical principles outlined in the Declaration of Helsinki for medical research involving human subjects and received approval from the relevant institutional ethics review board.

The samples were collected and categorized into two groups: non-prediabetic (NPD) (n = 40) and prediabetic (PD) (n = 40). Prediabetes was determined based on glycated hemoglobin (HbA1c) levels falling within the range of 39–46 mmol/mol (5.7–6.4%), in accordance with the criteria set forth by the American Diabetes Association (ADA) [90]. HbA1c measurements were conducted at King Edward VIII Hospital (KEH) using the Tosoh G8 HPLC Analyzer (Tosoh Bioscience, Inc., 3604 Horizon Drive, Suite 100, King of Prussia, PA, USA), a laboratory-based instrument certified by both NGSP and IFCC. The blood samples were then subjected to centrifugation (Eppendorf centrifuge 5403, Hamburg, Germany) at 4 °C, at a speed of 10,000× *g* for a duration of 15 min. Subsequently, plasma was extracted and stored at −80 °C in a Bio Ultra freezer (Snijers Scientific, Tilburg, Holland) until biochemical analysis.

### 4.2. Biochemical Analysis

Plasma adipsin, adiponectin, insulin, and MCP-1 concentrations were assessed using the Human Metabolic Hormone Magnetic Bead Panel protocols from the MILLIPLEX^®^ MAP Kit (Merck, Darmstadt, Germany), following the manufacturer’s guidelines. The concentrations of MILLIPLEX were detected using the Bio-Plex MAGPIX Multiplex Reader (Bio-Rad Laboratories Inc, Hercules, CA, USA) and quantified using the Bio-Plex Manager version 6.1 software. The plasma concentrations of the RAS components, namely renin, angiotensin-converting enzyme 1 (ACE1), angiotensin II (Ang II), angiotensin-converting enzyme 2 (ACE2), and angiotensin 1-7 (Ang 1-7), were determined using specialized immunoassays. Human ACE1 levels were quantified using the ACE1 ELISA Kit (ab108637) from Abcam (Cambridge, UK), whereas Ang II concentrations were measured using the Angiotensin II ELISA Kit (ab213803) from Abcam (Cambridge, UK). ACE2 levels were assessed using the angiotensin-converting enzyme 2 ELISA Kit (ab207325) from Abcam (Cambridge, UK), and Ang 1-7 was determined using the Angiotensin 1-7 ELISA Kit (ab213328) from Abcam (Cambridge, UK), according to the manufacturer’s instructions for optimal results.

### 4.3. Plasma Lipid Profile Determination (LDL and VLDL Calculation)

LDL and VLDL were determined using Friedewald equationCalculate LDL cholesterol using the Friedewald equation:LDL cholesterol (mg/dL) = total cholesterol (mg/dL) − HDL cholesterol (mg/dL) − (triglycerides (mg/dL)/5)Calculate VLDL cholesterol:VLDL cholesterol (mg/dL) = Triglycerides (mg/dL)/5Assay Sensitivities (minimum detectable concentrations, pg/mL)Link Between Lipid Profile and Insulin Resistance

Dyslipidaemia, characterized by elevated triglycerides and reduced HDL-C, is a hallmark of insulin resistance and is commonly observed in individuals with metabolic dysfunction [91]. Given this established link between lipid abnormalities and insulin sensitivity, further analysis was conducted to determine the degree of insulin resistance using the HOMA-IR index.

Insulin Resistance Determination (HOMA-IR Calculation)

The Homeostatic Model Assessment for Insulin Resistance (HOMA-IR) was calculated using the following formula:HOMA-IR = (Fasting insulin [µU/mL] × Fasting glucose [mmol/L])/22.5

This index provides an estimate of insulin resistance, with higher values indicating reduced insulin sensitivity.Assay Sensitivities (minimum detectable concentrations, pg/mL)

As stated in the protocol, the Minimum Detectable Concentration (MinDC) was calculated using the MILLIPLEX Analyst 5.1. The true limits of detection for an assay were measured by mathematically determining what the empirical MinDC would be if an infinite number of standard concentrations were run for the assay under the same conditions (n = 8) (Table 3).

#### 4.3.1. Precision

As stated in the protocol, the intra-assay precision was generated from the mean of the %CV’s from eight reportable results across two different concentrations of analytes in a single assay. Inter-assay precision was calculated from the mean of the %CV’s across the two different concentrations of analytes across six different assays (Table 4).

#### 4.3.2. Statistical Analysis

All statistical analyses were performed using the GraphPad Prism version 8.0.2 software (GraphPad Software Inc., San Diego, CA, USA). The Shapiro–Wilk test was used to assess the normal distribution of the data. For parametric data, differences between the means of the independent non-prediabetic (NPD), prediabetic (PD), and type 2 diabetic (T2DM) groups were assessed using one-way analysis of variance (ANOVA). All non-parametric data were analyzed with the Mann–Whitney U test or log-transformed to meet the assumptions of normality to achieve normal distribution for the same statistical analysis and reported as mean and standard deviation (SD). The Kruskal–Wallis test was used to assess non-parametric data. A *p*-value of <0.05 was considered significant. The correlation between the groups was determined using Pearson’s correlation and is reported in Appendix A.

## 5. Conclusions

Given the upregulated ACE1/Ang II arm of RAS in PD, the results of the current study suggest that RAS is altered in prediabetes. The shift in RAS was directly proportional to the reduction in adipokines such as adiponectin and adipsin, in addition to the increase in the activation of inflammatory markers such as MCP-1. Therefore, the current study highlights the relationship between RAS and moderate hyperglycemia, as observed in PD, in association with adipocyte dysfunction. Although this study did not include anthropometric measures of adiposity, its focus on biochemical indicators provides meaningful insight into tissue-level dysregulation in prediabetes. Our findings reveal elevated markers of adipose tissue dysfunction in the PD group, suggesting early metabolic dysregulation. This is particularly relevant given the high burden of overweight and obesity in South Africa [92]. Mphasha et al. (2022) observed a substantial prevalence of overweight and obesity among diabetic patients and their first-degree relatives in Limpopo province [93]. Similarly, Sosibo et al. (2022) reported a high prevalence of prediabetes and its association with obesity across diverse South African ethnic groups [94]. These findings support our biochemical observations and suggest that adipose tissue dysfunction may play a critical role in the pathophysiology of prediabetes, thus underscoring the need to consider adiposity as a key factor in prediabetic pathophysiology. Future studies would benefit from incorporating direct measurements of adiposity to deepen understanding of the RAS–adipose tissue axis in metabolic disease.

This work provides an important foundation for identifying novel biomarkers and therapeutic targets in prediabetes, particularly those that may disrupt maladaptive RAS signalling and preserve adipose tissue function. Therefore, therapeutic pathways that target specific RAS components may be beneficial for preventing the progression of PD to T2DM and the development of comorbidities associated with T2DM, such as obesity, lipotoxicity, adiposity, and insulin resistance. Through this study, we envisage that more developments towards underpinning the role of RAS and its attenuation strategies in the conversion of prediabetes to overt diabetes will emerge.

### 5.1. Study Limitations

Although our study successfully investigated the interplay between RAS components, glucose levels, and biochemical alterations in adipose tissue, certain enhancements could have further strengthened its comprehensiveness. Notably, the inclusion of additional parameters, such as inflammatory markers along with RAS activity, rather than solely focusing on RAS concentration levels, would have provided a more comprehensive understanding of the underlying mechanisms involved in metabolic dysregulation.

### 5.2. Future Studies

This study was limited by its retrospective design, which precluded the collection of important anthropometric measurements such as body mass index (BMI), waist circumference, and blood pressure. The absence of these variables limited the ability to explore potential associations between metabolic risk factors and clinical outcomes. Additionally, while the current sample size was sufficient for preliminary analyses, a larger cohort would enhance statistical power and improve the generalisability of the findings. Future studies are planned to adopt a longitudinal design, allowing for the collection of comprehensive anthropometric, clinical, and biochemical measurements

### 5.3. Patents

The study protocol was approved by the University of Kwa Zulu-Natal Biomedical Research Ethics Committee (BREC) (reference no. BE266/19), and the Provincial Health Research Committee of the KZN Department of Health. All the participants provided informed consent.

## Figures and Tables

**Figure 1 ijms-26-05517-f001:**
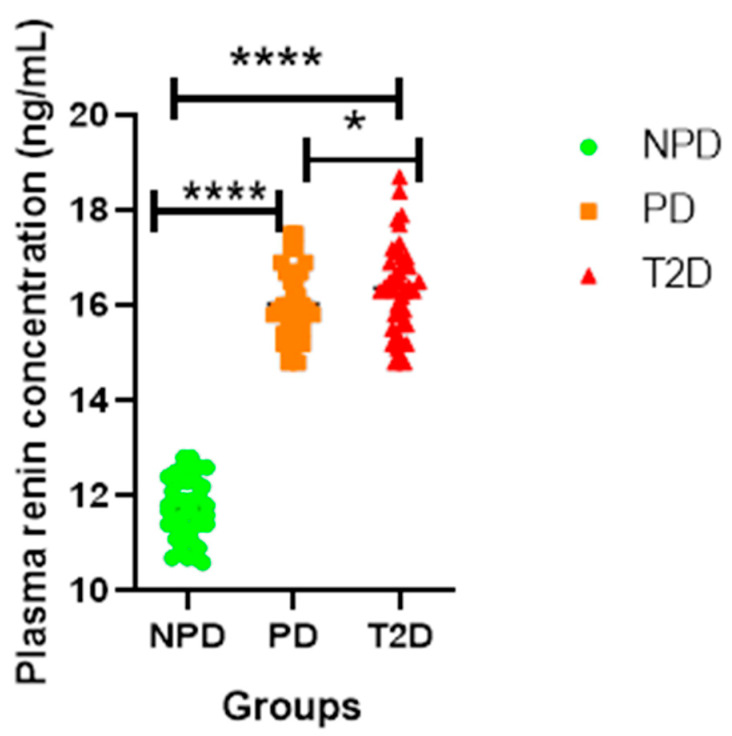
Renin plasma concentration in non-prediabetic (NPD), prediabetic (PD), and type 2 diabetes mellitus (T2DM) participants (n = 40, per group). Values are expressed as mean ± SD. **** *p* = <0.0001 denotes a comparison of NPD with PD. **** *p* = <0.0001 denotes comparison of NPD with T2D and * *p* = 0.0317 denotes comparison of PD with T2D.

**Figure 2 ijms-26-05517-f002:**
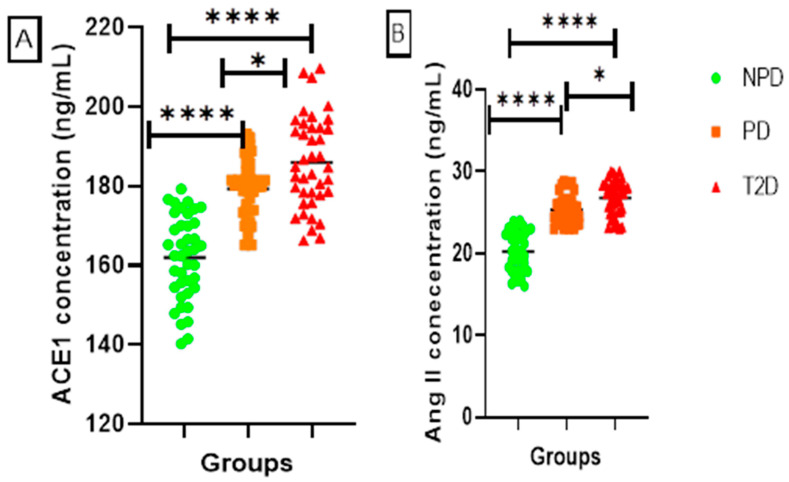
(**A**) ACE1 and (**B**) Ang II plasma concentration in non-prediabetic (NPD) and prediabetic (PD) participants (n = 40, per group). Values are expressed as mean ± SD. (**A**) **** *p* = <0.0001 denotes comparison between NPD and PD in addition to the comparison between T2D with NPD and PD, whilst * *p* = 0.026 denotes comparison of PD with T2D. (**B**) **** *p* = <0.0001 denotes comparison between NPD and PD in addition to the comparison of NPD with PD and TPD with T2D, whist * *p* = 0.0138 denotes comparison of PD with T2D.

**Figure 3 ijms-26-05517-f003:**
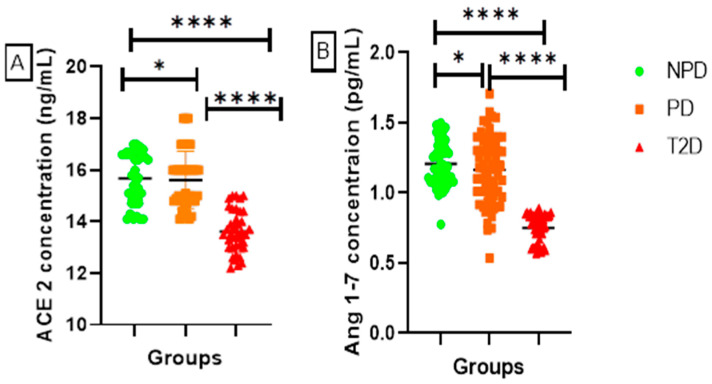
(**A**) ACE2 and (**B**) Ang 1-7 plasma concentration in non-prediabetic (NPD) and prediabetic (PD) and type 2 diabetes (T2D) participants (n = 40, per group). Values are expressed as mean ± SD. (**A**) * *p* = 0.0431 denotes comparison of NPD with PD and **** *p* = <0.0001 denotes comparison between PD and T2D and NPD with T2D, and (**B**) * *p* = 0.0215 denotes comparison between NPD and PD, whilst **** *p* = <0.0001 denotes comparison of NPD with PD and T2D in addition to a comparison between PD and T2D.

**Figure 4 ijms-26-05517-f004:**
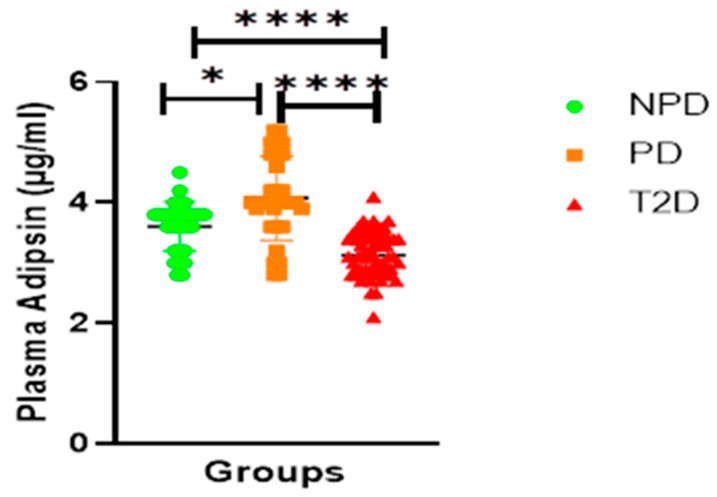
Plasma adipsin concentration in non-prediabetic (NPD), prediabetic (PD), and type 2 diabetes (T2D) participants (n = 40, per group). Values are expressed as mean ± SD. * *p* = 0.0321, **** *p* = <0.0001 denotes comparison of NPD with PD, PD with T2D and T2D with NPD.

**Figure 5 ijms-26-05517-f005:**
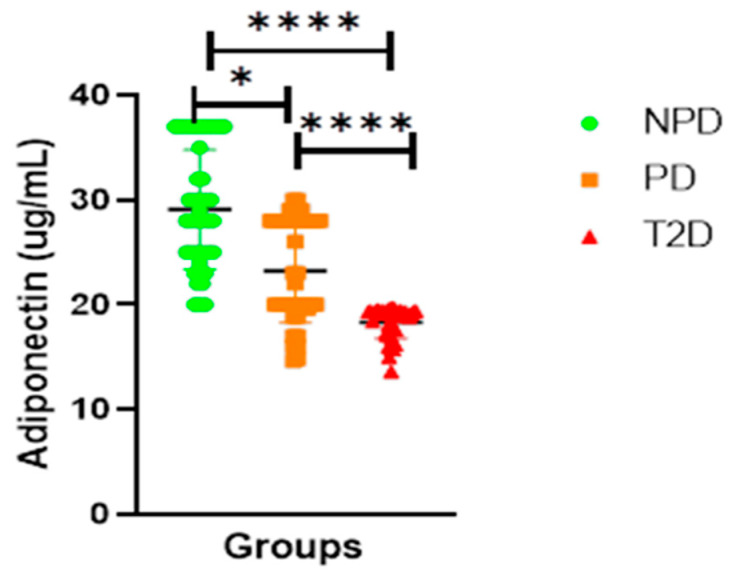
Adiponectin plasma concentration in non-prediabetic (NPD), prediabetic (PD), and type 2 diabetes (T2D) participants (n = 40, per group). Values are expressed as mean ± SD. * *p* = 0.0463 and **** *p* = <0.0001 denotes comparison of NPD with PD, PD with T2D and T2D with NPD.

**Figure 6 ijms-26-05517-f006:**
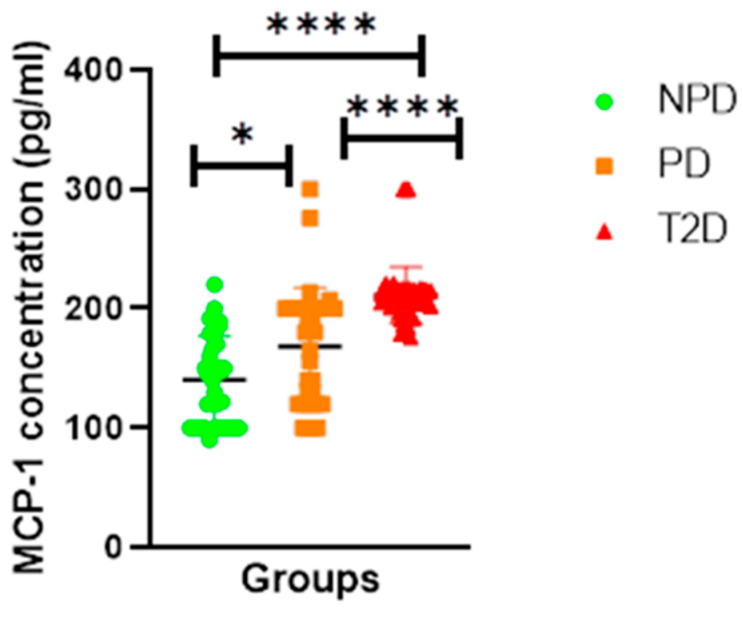
MCP-1 plasma concentration in non-prediabetic (NPD), prediabetic (PD), and type 2 diabetes (T2D) participants (n = 40, per group). Values are expressed as mean ± SD. * *p* = 0.0313 and **** *p* = <0.0001 denotes comparison of NPD with PD, PD with T2D and T2D with NPD.

**Table 1 ijms-26-05517-t001:** The characteristics of the patients included in this study.

	NPD (n = 40)	PD (n = 40)	T2D (n = 40)	Total (n = 120)	Overall *p*	NPD vs. PD	NPD vs. T2D	PD vs. T2D
Age (years)	36.95 ± 6.26	37.28 ± 6.51	40.30 ± 4.96	120	*p* = 0.03	*p* = 0.77	*p* = 0.02	*p* = 0.04
Gender								
Males, n (%)	18 (45.00)	20 (50.00)	19 (47.50)	57				
Females, n (%)	22 (55.00)	20 (50.00)	21 (52.50)	63				
FBG (mmol/L)	5.33 ± 0.65	6.88 ± 0.63	12.47 ± 4.54	120	*p* < 0.0001	*p* < 0.0001	*p* < 0.0001	*p* < 0.0001
TGs (mmol/L)	1.56 ± 0.14	1.99 ± 0.25	2.76 ± 1.12	120	*p* < 0.0001	*p* < 0.0001	*p* < 0.0001	*p* = 0.01
Total cholesterol (mmol/L)	4.42 ± 0.39	5.53 ± 0.48	5.59 ± 0.45	120	*p* < 0.0001	*p* < 0.0001	*p* < 0.0001	*p* = 0.61
HDL (mmol/L)	1.01 ± 0.11	0.67 ± 0.11	0.50 ± 0.06	120	*p* < 0.0001	*p* < 0.0001	*p* < 0.0001	*p* < 0.0001
LDL (mg/dL)	107.20 ± 16.75	145.70 ± 13.82	156.20 ± 22.29	120	*p* < 0.0001	*p* < 0.0001	*p* < 0.0001	*p* = 0.05
VLDL (mmol/L)	0.72 ± 0.06	0.91 ± 0.10	1.23 ± 0.50	120	*p* < 0.0001	*p* < 0.0001	*p* < 0.0001	*p* = 0.02

FBG, fasting blood glucose; TG, TG triglycerides; HDL, high-density lipoprotein; LDL, low-density lipoprotein; VLDL, very low-density lipoprotein; Nonparametric Kruskal–Wallis test.

**Table 2 ijms-26-05517-t002:** Glycated hemoglobin and Homeostatic Model Assessment of insulin resistance (HOMA-IR) concentration in non-prediabetic (NPD), prediabetic participants (PD), and type 2 diabetes mellitus (T2DM) (n = 40 per group). HbA1c and HOMA-IR values are expressed as mean ± SD.

Groups	NPD	PD	T2DM
HbA1c %	4.9 ± 0.4	5.9 ± 0.6	7.3 ± 1.1
HOMA-IR	2.1 ± 0.4	3.9 ± 0.8	5.9 ± 1.4
*p*-values		<0.0001	<0.0001

**Table 3 ijms-26-05517-t003:** Assay sensitivities minimum detectable concentrations of Insulin, MCP-1, Adiponectin and Adipsin.

Analyte	MinDC (pg/mL)	MinDC + 2SD (pg/mL)
Insulin	87	160
MCP-1	14	14
Adiponectin	11	21
Adipsin	4.8	10

**Table 4 ijms-26-05517-t004:** Intra-assay precision concentrations of Insulin, MCP-1, Adiponectin and Adipsin.

Analyte	Intra-Assay %CV	Inter-Assay %CV
Insulin	<10	<15
MCP-1	<10	<15
Adiponectin	<10	<15
Adipsin	<10	<10

## Data Availability

The data supporting the findings of this study are openly available in the Zenodo repository at https://doi.org/10.5281/zenodo.14169610. Data have been made available in accordance with the MDPI Available Policy, and all relevant materials supporting this work can be freely accessed through the provided link.

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
