# Peer review of "Increased Plasma Levels of ACE and Ang II in Prediabetes May Contribute to Adipose Tissue Dysfunction"

_ijms, 2025, doi:10.3390/ijms26125517_

Round 1
Reviewer 1 Report
Comments and Suggestions for Authors
There exist numerous relationships between the RAS-Angotensin system and adipocytes. The interactions in individuals with diagnosed diabetes are well known. The authors attempted to delineate relationships among patients with intermediate hyperglycemia (prediabetes) and the RAS-ACE-Ang system. We all know that "prediabetes" is a somewhat artifical designation on the continuum of impairment of normal glucose control. Nonetheless it is valuable to address this issue since we are subject to these artificial demarcations. They present a very small number of subjects in the three classes of normal glycemia, diagnosed type 2 diabetes and prediabetes. Therefore their presentation can only be viewed as a preliminary pilot study. Yet, it certainly is a worthwhile attempt. What is missing is a phenotypic characterization of the subjects, notably BMI, waist circumference, and blood pressure data which is not presented. A particular question is whether these findings relate to glucose levels or are simply a function of obesity. In terms of discussion, much more is needed in terms of background, particularly of prior studies of type 2 diabetes patients. What one would like to see is correlation analysis, clearly impossible with such small patient numbers. Again this is a pilot study.
Reviewer 2 Report
Comments and Suggestions for Authors
Three study groups are mentioned in the abstract, statistical analysis and results, while only two groups (non-prediabetic and prediabetic) are mentioned in the materials and methods section. Therefore, it is important to correct the number of study groups.
In addition, the section on the analysis of lipoproteins (HDL, LDL and VLDL) mentioned in the materials and methods section is confusing for the reader and it is suggested to improve its synthesis and to clearly link why the information from the tables of glycated hemoglobin and other parameters of insulin resistance is necessary in this section.
Authors are encouraged to improve the quality of their figures.
Round 2
Reviewer 1 Report
Comments and Suggestions for Authors
This is a useful pilot study considering the relationship between the renin angiotensin system and markers of adipocyte function. What is clearly missing is a relationship to adiposity. The authors carried out a laboratory study without adequate characterization of their subjects with respect to BMI, waist circumference, and blood pressure. The label "prediabetes" begs the question as to the adipose tissue burden in their subjects. Nonetheless their study has value as an initial attempt which must be clearly described as exploratory.
Round 3
Reviewer 1 Report
Comments and Suggestions for Authors
I have tried to convey to the authors that their attempt to assess renin angiotensin system response in a retrospective study of patients with prediabetes without any knowledge of the degree of adiposity of the patients is of interest but is harmed by the lack of a control group of adipose patients who do not meet the diagnostic criteria to be labelled as either having diabetes or not having diabetes. So we do not know if their findings have any relationship to glucose-insulin dynamics or purely to adipose tissue relationships.
Comments on the Quality of English LanguageEnglish is fine.
Round 4
Reviewer 1 Report
Comments and Suggestions for Authors
The authors have accumulated useful data. The flaw in their research design is that the study did not collect data on patient adiposity. If they wish to publish, then they should state this openly in the introduction and then explain in the conclusion. Having said this, it is highly likely that prediabetes and diagnosed type 2 diabetes patients were all obese. to strengthen their presentation, they might reference the percentage obesity in their general clinics which I am guessing is very high.
Comments on the Quality of English LanguageThe English is fine
Round 5
Reviewer 1 Report
Comments and Suggestions for Authors
The authors have now addressed the principal weakness of their study. They did not account for adipose tissue in their population. Yet, their biochemical data is valuable. To strengthen their presentation would include references obesity and overweight in the South African population with prediabetes:
Sosibo AM, Mzimela NC, Ngubane PS, Khathi A. Prevalence and correlates of pre-diabetes in adults of mixed ethnicities in the South African population: A systematic review and meta-analysis. PLoS One. 2022 Nov 29;17(11):e0278347.
Mphasha MH, Skaal L, Mothiba TM. Prevalence of overweight and obesity amongst patients with diabetes and their non-diabetic family members in Senwabarwana, Limpopo province, South Africa. S Afr Fam Pract (2004). 2022 May 25;64(1):e1-e7
The incorporation of these references will strengthen their argument.
